# Assessment of Landscape Retention Water Capacity and Hydrological Balance in Traditional Agricultural Landscape (Model Area Liptovská Teplička Settlements, Slovakia)

**Zdena Krnáčová** [1], **Pavol Kenderessy** [1], **Juraj Hreško** [2,*], **Daniel Kubínsky** [3] and **Marta Dobrovodská** [1]

1   Institute of Landscape Ecology, Slovak Academy of Sciences, Štefánikova 3, P.O. Box 254,
    814 99 Bratislava, Slovakia; zdena.krnacova@savba.sk (Z.K.); pavol.kenderessy@savba.sk (P.K.);
    marta.dobrovodska@savba.sk (M.D.)
2   Department of Ecology and Environmental Sciences, Faculty of Natural Sciences,
    Constantine the Philosopher University, A. Hlinku 1, 949 01 Nitra, Slovakia
3   Slovak Environment Agency, Tajovského 28, 975 09 Banská Bystrica, Slovakia; daniel.kubinsky@sazp.sk
*   Correspondence: jhresko@ukf.sk

**Abstract:** The hydration potential of a landscape is an increasingly important attribute in a time of advancing climate change, making its assessment also a matter of some urgency. This study used the landscape ecological approach involving the hydrological balance, in which the soil water retention capacity (SWRC) and landscape water retention capacity (LWRC) are evaluated. To support our assessment of the water retention capacity in the landscape (LWRC), we used a synthetic interconnection of analytical vector layers of selected physical parameters of soil subtypes and secondary landscape structure (SLS) to create homogeneous polygons in the GIS Arc/Map10 computing environment. Selected abiotic and biotic attributes were assigned coefficients using a simple algorithm according to the authors, which were projected into landscape ecological complexes (LEC) in the GIS computer program in the Arc/Map10 program. We used hydrological balance calculations to specify the volumes of water retained in the landscape. The aim is to spatially estimate the retention capacity of the landscape, taking into account the current land use, including historical anti-erosion measures to reduce unwanted water runoff and soil erosion. Using zonal statistics, we achieved the following results. The part of the model area with very low or low LWCR represents 39.91% of the agricultural land used. We recorded a high LWCR on 17.69% of the area, with a predominance of meadows and cultizol cambis and cultizol fluvials. The calculation of the hydrological balance, which represents only 22.9% of atmospheric precipitation, also made a significant contribution to our knowledge of the LWRC.

**Keywords:** landscape water retention capacity (LWRC); soil water retention capacity (SWRC); landscape ecological approach; GIS tolls; critical hydric zones; zonal statistics; traditional agricultural landscape; hydrological balance

## 1. Introduction

Manifestations of climate change and ongoing land use change in the mountainous regions of the Western Carpathians determine the retention capacity of the soil and landscape, which, on the one hand, can manifest itself in hydrological drought and, on the other hand, can cause risks associated with extreme surface water runoff and floods. The latter leads to extensive property damage and,

unfortunately, loss of human lives. Therefore, research has long focused on the development, creation, and refinement of hydrological models, wherein the potential of the land to retain water is explored, based on which effective measures are proposed to alleviate this unfavorable condition [1,2].

The landscape water retention capacity (LWRC) is the ability of the landscape to retain water after it has fallen as rain, releasing it later to complete the water cycle. The rain which falls on such an area is taken up by the vegetation or the water bodies and recharges the groundwater. Where soils have a thicker humus layer, the retention areas have a higher ability to prevent large-scale landslides as well as floods, whose occurrence after increased rainfall is becoming more frequent [3].

An essential step for dealing with integrated river basin management is adoption of Directive 2000/60/EC. Its purpose is to establish an integrated hydrological framework as EU policy, in order to protect the physical and biological integrity of water systems and to reduce negative pressures on drinking water sources [4] (Directive 2000/60/EC of the European Parliament and of the Council on water policy).

More recently, IWRM has been expanded by integrating all water resources into the framework, not only water in streams and reservoirs as was the original concept for IRWM. This extended approach to IRWM is now being advocated and is often referred to as including "blue" and "green" water, making the distinction between free water in streams and reservoirs and water available in unsaturated soils to be used by the vegetation or crops [5].

Water retention in the country is becoming more and more relevant in the context of global change. In connection with water retention, several adaptation strategies for IMP have been developed [6,7], which mention the possibilities of rational water management and retention in the country. However, it is worth mentioning that a number of publications on the issue have dealt with land use optimization, sustainable development, soil protection and soil conditions, stream hydrology, forest and non-forest landscape characteristics, nature and landscape protection, and all elements and components of the landscape with which water is interconnected as an integrating element; and which provide the basic analytical data for IMP (Integrated Management Plan) [5,8,9].

The most significant research so far on water retention in the countryside can be found in studies by [10]. Knowledge of the patterns of runoff formation and analysis of runoff in river basins with different natural conditions show that, out of all the characteristics of river basins, it is the geological structure of the area that has the greatest influence on runoff, and also to a large extent affects the basic properties of the soil.

A detailed analysis of flood events that occurred in Slovakia in the years 1996 to 2006 [10] showed that floods mostly occurred in areas with less permeable bedrock (Paleogene flysch) located in areas with reduced soil permeability. Most of them occurred in small river basins with areas of 300 km$^2$ or less, and the most common cause was short, intense rains (so-called "flash floods").

Infiltrated water feeds springs and streams without the formation of surface runoff in medium and non-extreme conditions. In the case of extreme precipitation, that is, long-duration, high-intensity precipitation, surface runoff is also associated with subsurface runoff [11].

During high-intensity torrential rainfalls in the upper parts of mountain basins with relatively steep slopes, local floodwaters are exacerbated by the shallowness of the soil profile (and the relatively small retention capacity that results) and the occurrence of a low-permeability rock layer at a depth of less than one meter. After the saturation of soil profile with water, subsurface and subsequently surface runoff occur, which can cause erosion and floods. These unfavorable soil properties cannot be significantly managed in terms of formation of local floods, and thus the management of soil cover properties does not lead to a reduction of floods [11].

We considered the landscape ecological approach in planning our study. The ecological approach to the evaluation of the landscape and its attributes is based on the principles of the LANDEP (Landscape Ecological Planning) methodology by Ružička and Miklós [12]. This methodology has gained significant international recognition by being included in the international document AGENDA

21 (adopted at the World Summit in Rio de Janeiro in 1992) among the recommended methodologies for ensuring an integrated approach to landscape planning and management.

Landscape ecological approaches, in the form of landscape ecological plans (LEP), are gradually being integrated into spatial planning and several sectoral planning processes (e.g., sustainable management programs for protected nature areas, river basin management plans, and land management projects) [13]. LEP are already enshrined in legislation, namely in the Act of the National Council of the Slovak Republic no. 237/2000 Coll., amending and supplementing Act no. 50/1976 Coll. on Spatial Planning and Building Regulations (Building Act), which came into force on the 1 August 2000.

Landscapes with low soil retention capacity and inappropriate land use are identified as critical hydric zones, where there is a high presumption of high runoff of soil and groundwater, and require alleviation of this unfavorable state of aridization and gradual degradation of soils and land by changing land use. Slowing down the runoff of rainwater will create the conditions for its soaking into soils and thus allow the development or restoration of vital functions of the landscape. For more accurate estimates of the volume of water retained in the country, we also calculated the hydrological balance. Hydrological balance, also termed the water regime, occurs relatively often in research in the field of ecology, environmental studies, and, especially, hydrology. There are many ways and methodologies to determine it at a given location. Some are based on accurate field measurements in a given location and subsequent accurate mathematical modelling [14–16]. Other procedures are based on existing data with the application of expert analysis to supplement and refine the selected parameters. The hydrological balance is always related to a certain time period (daily, monthly, annual, or multiannual) [17,18].

However, it is worth mentioning a number of publications on the issue which have dealt with land use optimization, sustainable development, soil protection and soil conditions, stream hydrology, forest and non-forest landscape characteristics, nature and landscape protection, and all elements and components of the landscape with which water is interconnected as an integrating element; and which provide the basic analytical data for IMP [7,19–23].

In our work, we have selected the following methodologies as appropriate to the scientific questions we wish to address.

1. We expect that the study area will have a medium-to-high LWRC, as calculated on the basis of mapped and measured attributes of abiotic conditions and detailed mapping of the current use of the agrarian part of the cadastral area Liptovská Teplička; we shall verify the LWRC calculations by selected methodological procedures;

2. We intend to use hydrological balance methods (empirically based procedures for estimating the amount of surface runoff) to determine the real volumes of water retained in the country, the amount of runoff from the territory, and the amount of evapotranspiration.

Determining LWRC values is a very difficult issue that is relevant not only to the hydrological cycle and water management, but also to other issues relating to the environment (e.g., landscape ecology, ecosystem services, geotechnics, legislation, the interests of municipal government, or property rights.) For these reasons, we have tried to estimate LWRC values using methods based on the principles of an integrated, landscape-ecological approach in combination with a hydrological balance. The connection of the landscape-ecological approach (based on the LANDEP methodology) with the hydrological balance was used for the first time. The result of landscape ecological outputs is the determination of the country's potential for water retention capacity, but realistic estimates of the volume of water retained in the country cannot be determined. For this reason, we used an additional methodology of hydrological balance, where attributes of both the land and the land use enter, but in a very general level, the focus is on measured climatic data, evapotranspiration, from which it is possible to derive real volumes of retained water within the allocated landscape-ecological complexes (LEC). The hydrology itself, which has so far used climatic data, evapotranspiration and flow in the watercourse, is gradually developing and using the attributes of the soil and landscape, the so-called ecohydrology.

## 2. Study Area

The landscape of Liptovská Teplička is a mountain type of settlement, located in the erosion-denudation furrow of the foothills of the Low Tatras in the Čierny Váh basin. According to the authors of [1], the area is situated in a representative geoecosystem of polygenic hills and subdivided pediments, originally with fir–spruce forests, within the north-east (NE) part of the geological region of the Low Tatras. The total area of the cadastral area, according to the database of the Statistical Office of Slovakia (SO SR), is 9845.8 ha. In the study, we evaluated the agrarian use landscape, which occupies an area of 1812.48 ha.

The dominant features of the landscape of the Liptovská Teplička mountain settlement study area are, firstly, the use of a suitable valley location for the creation and development of settlements in an open erosion furrow in front of the alpine landscape of the Low Tatras (Figures 1 and 2) and, secondly, the land use methods, which are based on a narrow-strip field system. In many localities, a system of slope stabilization and anti-erosion measures has been created, which also fulfils other environmental and ecological functions.

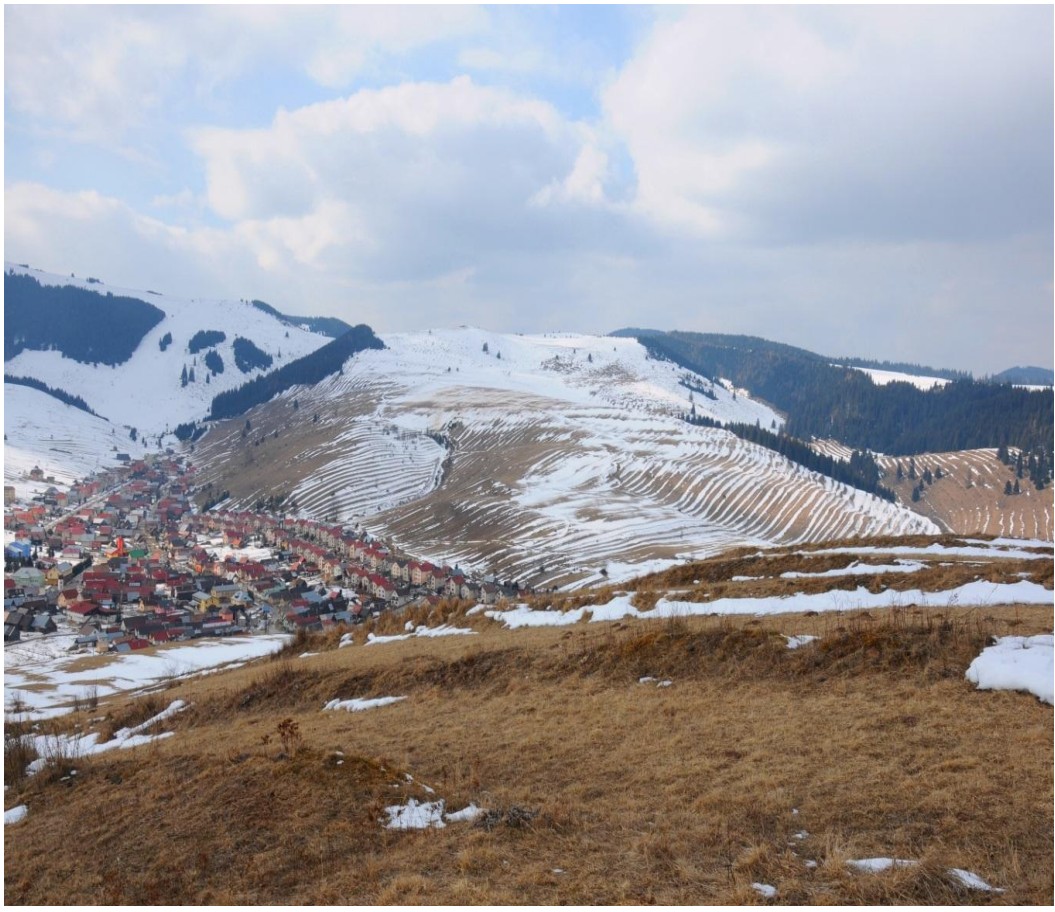

**Figure 1.** The traditional way of using the agrarian landscape with narrow-strip fields and plots, with the orientation of the ramparts in the direction of the slopes, photographed at the end of the winter season. Source: J. Hreško, 6.3. 2010.

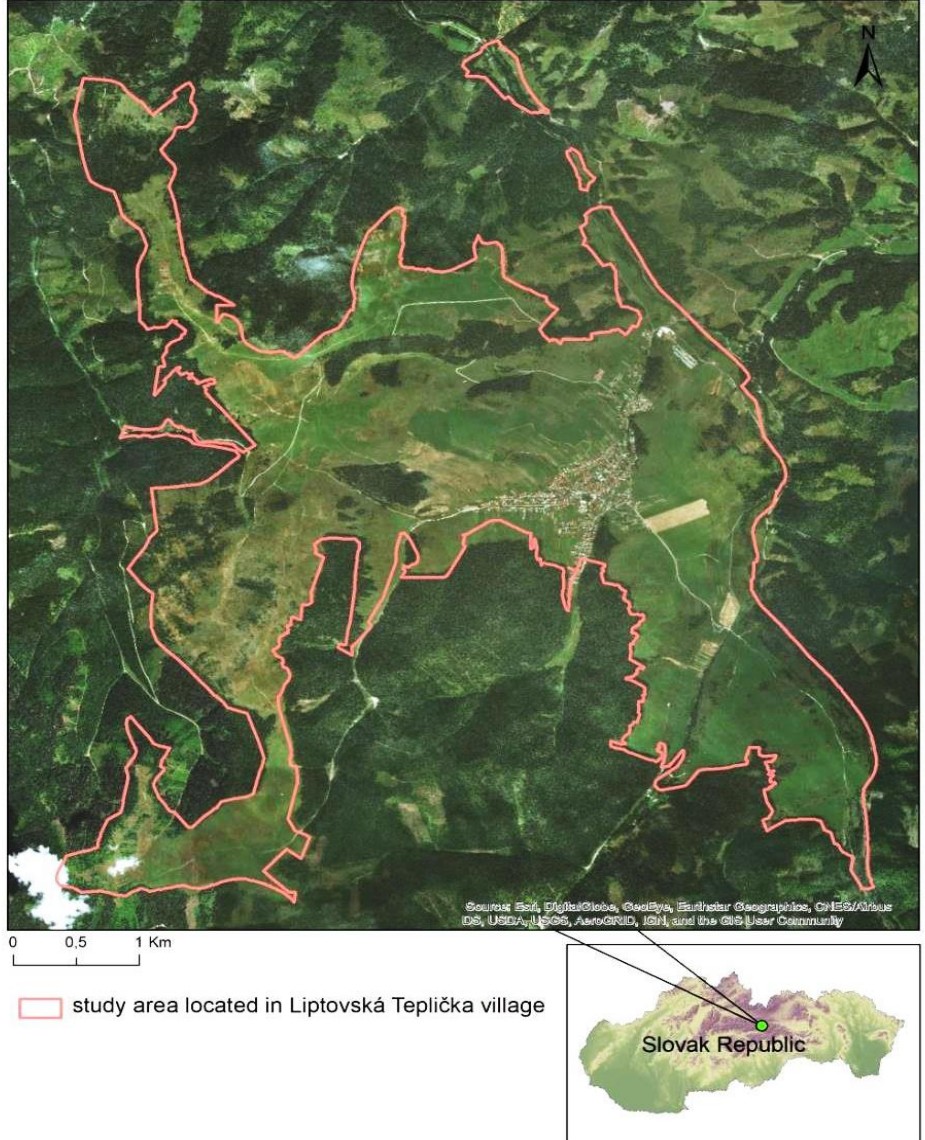

**Figure 2.** Geographical situation of the study area of Liptovská Teplička cadastre. Source: Base map, Esri, 2020.

The predominant way of using agrarian land is the extensive and intensive meadow and pasture form. The area belongs to two climatic districts, namely, to a cold mountain district with an average temperature in July of ≥10 °C and <12 °C and to a mild cold district with average July temperatures of ≥12 °C and <16 °C. The annual average total precipitation is 800 to 1000 mm, with 120 to 140 days a year having precipitation of 1 mm or more [24]. The course of annual precipitation and temperatures is shown in Figure 3. The measured climatic data are the basis for estimating the calculation of water retention in this agrarian landscape (Figure 3).

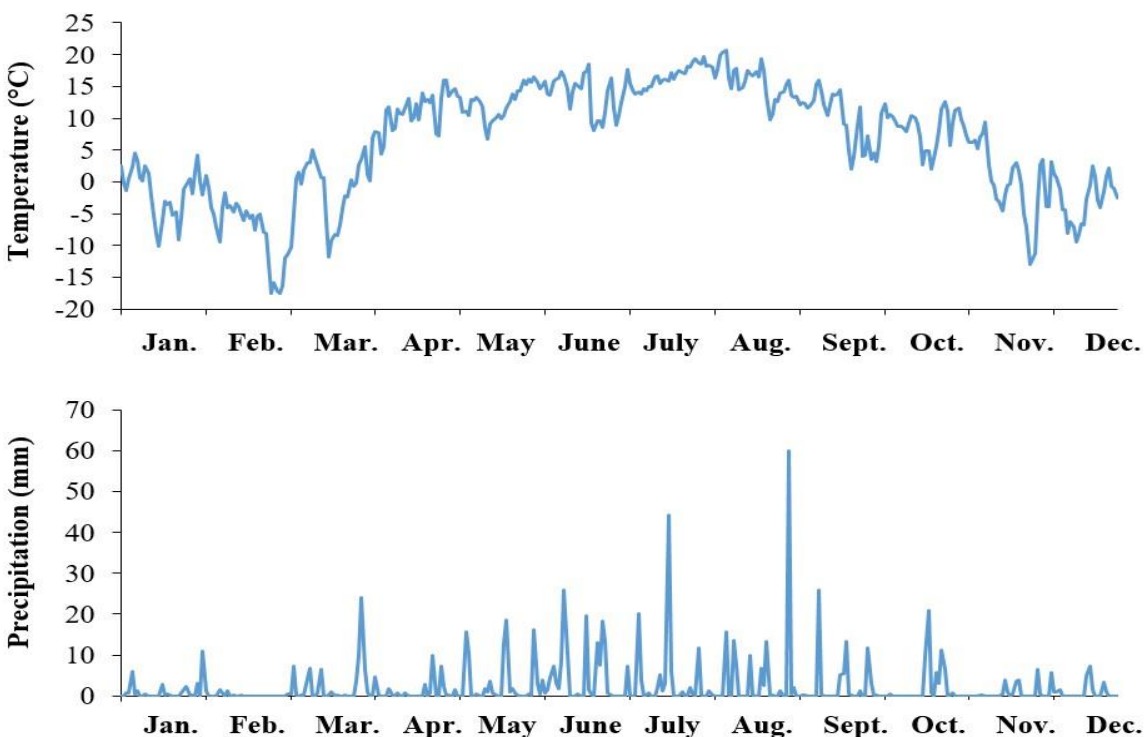

**Figure 3.** Temperature (°C) and precipitation variability (mm) per year 2018. The rainfall data were collected using an automatic meteorological station installed in the central part of the study area.

## 3. Methodological Approach

The ability of soil to retain water in its profile is among the most important soil functions. This ability is expressed as the water storage capacity or retention ability of the soil, and is affected especially by the physical properties of the soil. These are determined mainly by the granularity, structure, and parameters of the soil subtype. The soil water retention capacity (SWRC) is also significantly determined by the morphometric relief parameters. Water storage capacity together with infiltration speed determine the resistance of the environment to surface runoff or water stagnation at the surface of soils after torrential or heavy rains. No less important, however, are the use to which the landscape is put and the hydric properties of the elements of the secondary landscape structure. In the methodological section, we present the procedure for evaluation of the total LWRC of the agrarian part of the cadastral area of the village Liptovská Teplička.

### 3.1. Soil Water Retention Capacity (SWRC)

The soil water retention capacity can be expressed by the hydrolimits of the field capacity (FWC) The field capacity is a hydrolimit expressing limits put on the water content between gravitation and capillary action, and corresponds to a pressure of 2.0–2.9 pF1 [25]. Given the fact that the direct measurement of hydrological soil capacities is very difficult, statistically expressed pedotransfer functions (PTF) are currently used for the indirect estimation of hydrolimits [26]. The apparent correlation between $\Theta(h)$, $K(h)2$, and the content of individual soil grain-size fractions led to the formulation of an empirical model—the so-called pedotransfer function (PTF)—which is correlated to easily measured soil characteristics (granularity, specific weight, humus content, etc.) and hydrophysical soil characteristics [27–33].

In this work, the soil water retention capacity (SWRC) was assessed according to [33]. Their methodology was based on the synthesis of basic analytical layers such as soil subtypes and their attributes (such as grain size distribution and rock content), parent material, and morphometric parameters such as slope inclination. The spatial synthesis was performed using geoprocessing tools

in GIS. The results represent soil–substrate complexes, abiocomplexes (ABC), as homogeneous spatial units characterized by specific combinations of input parameters. The high diversity of resulting units arose mainly because of the highly complex geological-relief conditions of the study area, which, together with climatic, hydrological, and vegetation factors, strongly differentiated the soil cover and its characteristics.

According to the algorithm for quantification and calculation of the cumulative index for estimating the retention capacity of soil–substrate complexes [33], we determined the capacity of retained water in the soil profile and expressed how the volume of the water resources presented (derived resources from FVC (mm). The output data were reclassified into 10 categories according to their cumulative SWRC index values).

Input Analytical Data for SWRC Determination

- Database of vector layers of geological conditions according to the Regional Geological Map (at a scale of 1: 50,000) of the Low Tatras and Slovak [34].

Database of vector layers and classification of soil–substrate units (HPJ) according to the IUSS (International Union of Soil Science) International soil classification system for naming soils and creating legends for soil maps World Soil Resources Reports No. 106. FAO, Rome, 2015
Working Group WRB (World Reference Base), 2015. World Reference Base for Soil Resources 2014, update Morphogenetic Classification System of Soils of Slovakia [35] (S: 1:10,000).

- Slope parameter: inclination, from digitálny model reliéfu (DMR) data, with a resolution of 10 × 10 m.
- Database of vector layers of grain size and soil rock content [36] (S: 1:10,000).

*3.2. Landscape Water Retention Capacity (LWRC) Assessment*

The next step was to map the secondary landscape structure (SLS) and the occurrence of historical structures of the agrarian landscape. The landcover was mapped using the combined methods of visual interpretation of aerial photographs (s.r.o. Eurosense/Geodis, 2010) and field mapping. The land cover data were classified using the Corine Land Cover classification. In total, there were 26 classes identified in the study area (Figure 4). An adjusted weighting coefficient was assigned to the elements of the SLS according to the work of [2] in terms of their significance for water retention capacity. The historical structures of the agrarian landscape were identified according to [36] and verified during a field survey in the years 2015–2017. These structures were classified according to their orientation with respect to contour lines. This information was then used to assign the structures' respective water retention significance coefficient values. These data served as input databases for the synthesis of landscape ecological complexes (LEC) for determining the landscape's capacity for retention of water. This allows us to consider the presence of present-day land use and the amount of surface- and groundwater runoff when estimating water retention. The spatial synthesis was performed using geoprocessing tools in ArcGIS. The output raster represents the digital model of landscape retention estimation, reclassified into 10 categories.

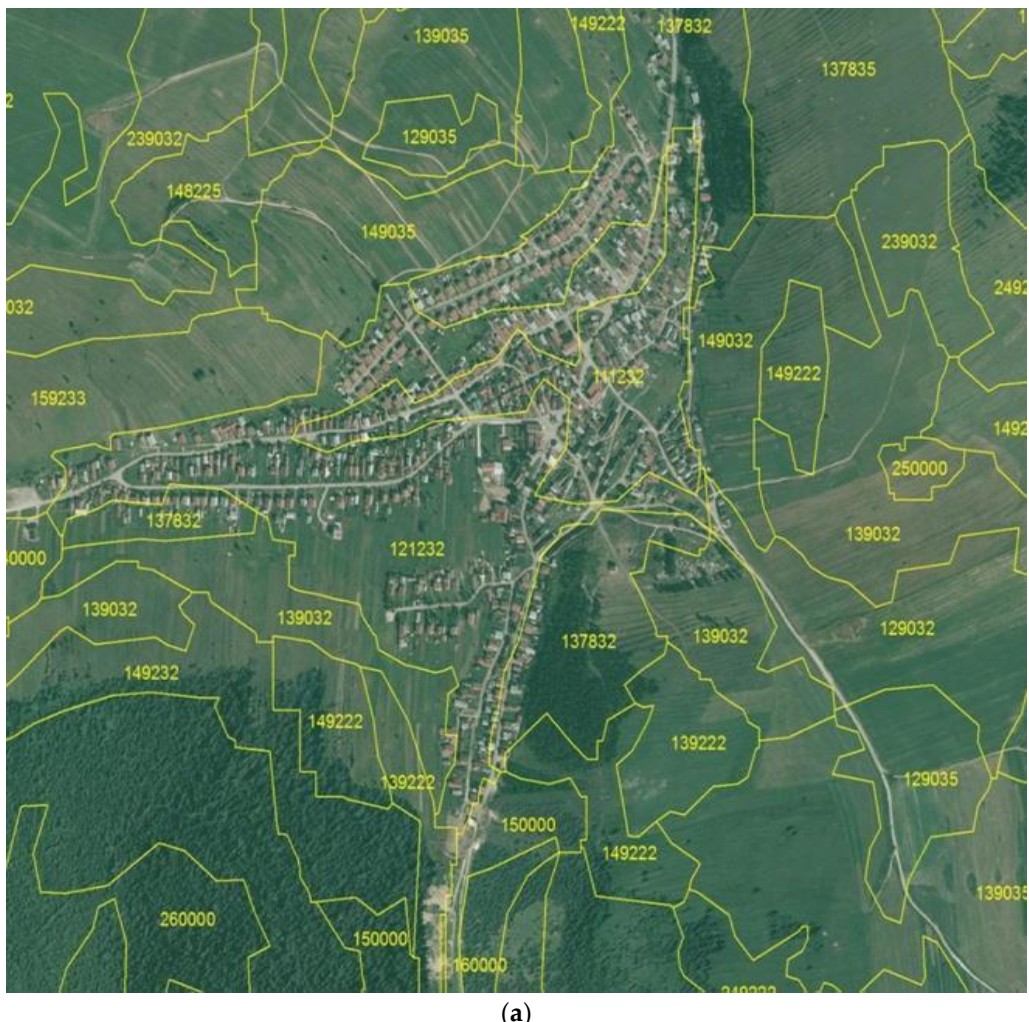

(**a**)

**Figure 4.** *Cont.*

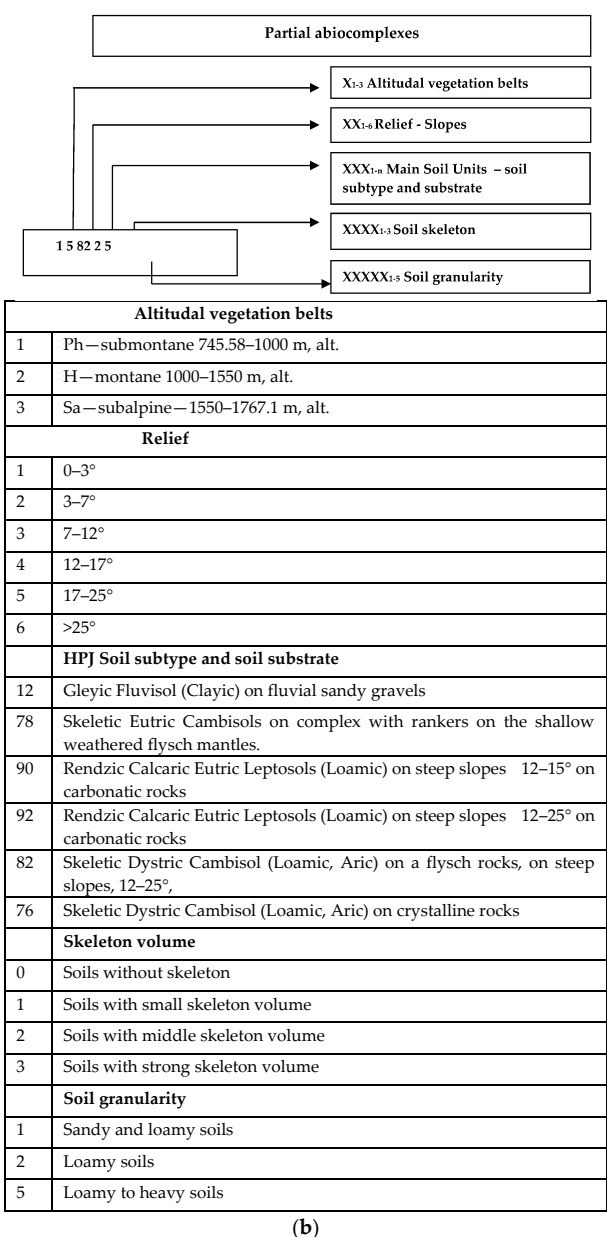

| Partial abiocomplexes | | |
|---|---|---|

| | | $X_{1-3}$ Altitudal vegetation belts |
| | | $XX_{1-6}$ Relief - Slopes |
| | | $XXX_{1-n}$ Main Soil Units – soil subtype and substrate |
| 1 5 82 2 5 | | $XXXX_{1-3}$ Soil skeleton |
| | | $XXXXX_{1-5}$ Soil granularity |

| Altitudal vegetation belts | |
|---|---|
| 1 | Ph—submontane 745.58–1000 m, alt. |
| 2 | H—montane 1000–1550 m, alt. |
| 3 | Sa—subalpine—1550–1767.1 m, alt. |
| **Relief** | |
| 1 | 0–3° |
| 2 | 3–7° |
| 3 | 7–12° |
| 4 | 12–17° |
| 5 | 17–25° |
| 6 | >25° |
| **HPJ Soil subtype and soil substrate** | |
| 12 | Gleyic Fluvisol (Clayic) on fluvial sandy gravels |
| 78 | Skeletic Eutric Cambisols on complex with rankers on the shallow weathered flysch mantles. |
| 90 | Rendzic Calcaric Eutric Leptosols (Loamic) on steep slopes 12–15° on carbonatic rocks |
| 92 | Rendzic Calcaric Eutric Leptosols (Loamic) on steep slopes 12–25° on carbonatic rocks |
| 82 | Skeletic Dystric Cambisol (Loamic, Aric) on a flysch rocks, on steep slopes, 12–25°, |
| 76 | Skeletic Dystric Cambisol (Loamic, Aric) on crystalline rocks |
| **Skeleton volume** | |
| 0 | Soils without skeleton |
| 1 | Soils with small skeleton volume |
| 2 | Soils with middle skeleton volume |
| 3 | Soils with strong skeleton volume |
| **Soil granularity** | |
| 1 | Sandy and loamy soils |
| 2 | Loamy soils |
| 5 | Loamy to heavy soils |

(**b**)

**Figure 4.** (**a**) Abiotic complexes (ABC) in model area. (**b**) Legend to Figure 4a.

Input Analytical Data for LWRC Calculation

- Analysis of the current land use from 2010, which was interpreted using data from basic maps of the Slovak Republic at a scale of 1:10,000 (from 1992–1993) and orthophotomaps at a scale of 1:5000 from 2002–2003 (Orthophotomap © Geodis Slovakia, s.r.o., 2003; Aerial photography and digital orthophotomap © Eurosense, sr o, 2003). These were verified by a field survey in the years 2015–2017. The secondary landscape structure was classified using the Corine Land Cover legend [37],
- Database of historical structures of the agrarian landscape (HSAL) [36]

$$K_{LWRC} = (K_{SWRC}) + (K_{SLSRC}) + (K_{HSALRC})$$

where the variables are as follows:

$K_{LWRC}$—coefficient of the landscape water retention capacity;

$K_{SWRC}$—coefficient of the soil water retention capacity (value of the cumulative index ranges from 0.1 to 11.0, see Table 1);

**Table 1.** Categories of cumulative soil water retention capacity (SWRC) index values.

| SWRC Category | Numerical Designation of Categories | The Range of Cumulative SWRC Index Values |
|---|---|---|
| Very low | 1 | 0.10–0.19 |
| | 2 | 0.20–2.28 |
| Low | 3 | 2.29–3.37 |
| | 4 | 3.38–4.56 |
| Medium | 5 | 4.57–5.55 |
| | 6 | 5.56–6.64 |
| High | 7 | 56.65–7.73 |
| | 8 | 7.74–8.82 |
| Very high | 9 | 8.83–9.91 |
| | 10 | 9.92–11.00 |

Source: Krnáčová, Hreško, Vlachovičová, 2016.

$K_{SLSRC}$—coefficient of secondary landscape structures with respect to retention capacity (RC) (value range 1.5–4, see Table 2);

**Table 2.** Secondary landscape structure classes and their respective retention coefficients in the study area of Liptovská Teplička.

| Element of Secondary Landscape Structure (SLS) CLC (Classification Code) | Retention Coefficient | SLS Area (ha) | SLS Area (%) |
|---|---|---|---|
| Built-up land area (1111) | - | 10.4 | 0.105 |
| Infrastructure (1221) | - | 32.18 | 0.33 |
| Landscaped areas up to 0.5 ha (14,113) | 2 | 1.49 | 0.02 |
| Cemeteries (14,121) | - | 1.30 | 0.01 |
| Home Gardens (1414) | 1.5 | 43.94 | 0.45 |
| Areas with predominantly grass sports areas (14,211) | 1.5 | 27.24 | 0.28 |
| Cottage settlements (14,221) | 1.5 | 0.31 | 0.00 |
| Other Recreation Areas (14,227) | - | 0.73 | 0.01 |
| Large-block fields with NWV up to 10% (2111) | - | 54.00 | 0.55 |
| Small-block fields (2112) | 1.5 | 4.96 | 0.05 |
| Intensively used meadows (23,111) | 3.0 | 196.75 | 2.00 |
| Extensively used meadows (23,112) | 4.0 | 276.75 | 2.81 |
| Abandoned meadows (degree of overgrowth of NWV up to 40%) (23,113) | 4.0 | 36.48 | 0.37 |
| Intensively used pastures (23,121) | 2.0 | 225.55 | 2.29 |
| Extensively used pastures (23,122) | 3.0 | 412.41 | 4.19 |
| Abandoned meadows (degree of overgrowth of NWV up to 40%) (23,123) | 4.0 | 56.89 | 0.58 |
| Mosaics of arable land and PGL (2415) | 3.0 | 49.17 | 0.50 |
| Crossings (3141) | 4.0 | 18.78 | 0.19 |
| Forest Nurseries (31,144) | 4.0 | | |
| Moorlands (322) | 4.0 | 3.30 | 0.03 |
| Groves (3241) | 4.0 | 11.19 | 0.11 |
| Groups of trees, shrubs (3242) | 4.0 | 5.92 | 0.06 |
| Woody riparian vegetation (3243) | 4.0 | 43.80 | 0.44 |
| Woodland line vegetation (3244) | 4.0 | 1.67 | 0.02 |
| Succession stages of woody plants (40 to 80% area coverage) on abandoned areas (3245) | 4.0 | 85.84 | 0.87 |
| Reed vegetation (3246) | 4.0 | 1.15 | 0.01 |
| Synanthropic vegetation with a share of woody plants of up to 40% (3251) | 3.0 | 1.84 | 0.02 |
| Total | | 1812.48 | 100.00 |

Source: Tužinský (2002), modified.

$K_{HSAL}$—coefficient of HSAL, that is, of significance of orientation ramparts and terraces of plots with respect to RC (value range 1–3, see Table 3);

**Table 3.** Overview of zonal statistics results and data for individual classes of retention (LWRC).

| LWRC Categories | Retention Class | MSU | MSU/m² | MSU/% | SLS /m² | SLS /% | Plot Orientation HSAL | Plot Orientation/m² | Plot Orientation /% |
|---|---|---|---|---|---|---|---|---|---|
| Very Low Retention (LWRC) (≤30 mm) per year | 1 (1.5–3.1) | 82 (RNa) | 141,910.77 | 0.92 | 167,140 | 0.92 | 1 | 6861.04 | 0.04 |
| | | 92 (RAa) | 25,229.93 | | | | | | |
| | | 76 (KMa) | 969.0 | | | | | | |
| | | 78 (KMa) | 8294.0 | | | | | | |
| | 2 (3.2–4.4) | 82 (RNa) | 308,849.0 | 3.57 | 648,712 | 3.57 | 1 | 97,123.08 | 0.54 |
| | | 90 (RAa) | 136,447.0 | | | | | | |
| | | 92 (RAa) | 194,156.0 | | | | | | |
| | | 76 (KMa) | 227,032.73 | | | | | | |
| | | 78 (KMa) | 196,999.56 | | | | | | |
| Low retention (LWRC) (30–70 mm) per year | 3 (4.5–5.1) | 82 (RNa) | 1,241,504.36 | 11.21 | 2034268 | 11.21 | 1 | 96,445.93 | 0.53 |
| | | 90 (RAa) | 190,898.36 | | | | | | |
| | | 92 (KMv) | 177,833.70 | | | | | | |
| | 4 (5.2–6.0) | 78 (KMa) | 461,596.52 | 24.13 | 4,380,043 | 24.13 | | | |
| | | 82 (RNa) | 777,899.37 | | | | 1 | 328,326.61 | 1.81 |
| | | 90 (RAa) | 737,463.11 | | | | 2 | 8076.10 | 0.04 |
| | | 92 (KMv) | 2403084.39 | | | | | | |
| Medium retention (LWRC) (70–110 mm) per year | 5 (6.1–7.0) | 78 (KMa) | 1298578.64 | 18.65 | 3,384,131 | 18.65 | | | |
| | | 82 (RNa) | 659407.44 | | | | 1 | 266,635.71 | 1.47 |
| | | 90 (RAa) | 1068059.66 | | | | 2 | 233,905.67 | 1.29 |
| | | 92 (RAa) | 26963.91 | | | | 3 | 30323.16 | 0.17 |
| | | 84 (KMg) | 88452.65 | | | | | | |
| | 6 (7.1–8.2) | 78 (KMa) | 1,793,815.64 | 24.36 | 4,421,114 | 24.36 | | | |
| | | 82 (KMa) | 358,105.05 | | | | 1 | 526,225.11 | 2.90 |
| | | 90 (RAa) | 2,018,434.20 | | | | 2 | 112,168.83 | 0.62 |
| | | 92 (RAa) | 124,535.52 | | | | 3 | 234,057.81 | 1.29 |
| | | 84 (KMg) | 126,224.01 | | | | | | |
| High retention (LWRC) (110–150 mm) per year | 7 (8.3–9.4) | 78 (KMa) | 588,546.26 | 10.91 | 1,980,066 | 10.91 | | | |
| | | 12 (FM$_G$) | 266,251.71 | | | | 1 | 229780.55 | 1.27 |
| | | 90 (RAa) | 793,256.22 | | | | 2 | 4219.81 | 0.23 |
| | | 92 (RAa) | 77,149.10 | | | | 3 | 125,721.83 | 0.69 |
| | | 84 (KMg) | 155,163.47 | | | | | | |
| | 8 (9.5–10.4) | 78 (KMa) | 5612.33 | 2.3 | 440,226 | 2.3 | 1 | 20624.68 | 0.11 |
| | | 90 (RAa) | 140,050.63 | | | | 2 | 81,255.49 | 0.45 |
| | | 84 (KMg) | 270,956.21 | | | | 3 | 40,983.87 | 0.23 |
| Very high retention (LWRC) (≥150 mm) per year | 9 (10.5–11.4) | 12 (FM$_G$) | 68,580.00 | 1.5 | 191,008 | 1.5 | 1 | 6582.19 | 0.79 |
| | | 90 (RAa) | 122,428.86 | | | | 2 | 289,296 | 0.04 |
| | | | | | | | 3 | 68,059.78 | 0.38 |
| | 10 (11.5–13.2) | 12 (FM$_G$) | 68,580.00 | 0.08 | 1065 | 0.08 | 2 | 9422.39 | 0.05 |
| | | 90 (RAa) | 122,428.86 | | | | 3 | 2426.01 | 0.01 |
| | Paved surfaces and isolated forests | | 487,669 | 2.69 | 487,669 | 2.69 | 0 | 0 | 0 |
| | Total | | 18,124,843 | | 18,124,843 | | | 2,298,492 | |

Legend: Main soils units (MSU) correlation according to The World Reference Base for Soil Resources (IUSS Working Group WRB, 2014): 12 (FM$_G$)—Gleyic Fluvisol (Clayic), 76 (KMa)—Skeletic Dystric Cambisol (Loamic, Aric) on crystalline rocks., 78 (KMa)—Skeletic Dystric Cambisol (Loamic, Aric) on flysch rocks, 82—(KNa) Skeletic Dystric Cambisol (Loamic, Aric) on a flysch rocks, on steep slopes, 12–25°, 84 (KMg)—Stagnic Cambisol (Loamic/Clayic) on steep slopes, 12–25°, 90 (RAa)—Rendzic Calcaric Eutric Leptosol (Loamic), and 92 (RAa)—Rendzic Calcaric Eutric Leptosols (Loamic) on steep slopes 12–25°. Plot orientation: 1—perpendicular to contour lines, 2—diagonal to contour lines, and 3—along the contour lines.

*RC*—retention capacity;

*HSAL*—historical structures of the agrarian landscape.

The result of this procedure is a raster map model of LWRC categorization in 10 categories, of which we will evaluate the potential for water retention in this study.

### 3.3. Hydrological Balance Assessment in the Tested Area

Hydrological balance (water balance and water budget) is an evaluation of increases and decreases in the amount of water and changes in its accumulation in the water body over a selected time interval. In its calculation, we subtract the volume of all water outflows from the water body from the volume of all inflows, and evaluate changes in water accumulation within that body with the selected calculation step (e.g., a day, a month, a year), usually over a longer period [38].

### 3.3.1. Input Analytical Data for the Calculation of the Hydrological Balance

- Value of the raster of potential evapotranspiration in mm, with respect to the use of land (all months during 2018, in GIS openable raster.) (Slovak Environmental Agency, MICROCOMP—computer system s.r.o.);
- Raster representation: DMR with a resolution of $10 \times 10$ m;
- Vector database of types of soil–substrate units (NPPC, 2014), converted into hydrological groups;
- Climate-data-monitoring database on precipitation and air temperatures (local hydrometeorological station, 2018 data by month);
- Software used to create the data layer: ArcGIS Desktop, version 10.6.1.

The basic formula for calculating the hydrological balance is to take the difference between water inputs and outputs in the area, and the resulting value is the amount of water reserves remained the area, expressed in mm, after a certain period:

$$\Delta S \ = P - (E + T) - Q,$$

where

$\Delta S$—changes in water reserves in the area (hydrological balance);

*P*—precipitation;

*E*, *T*—evaporation and transpiration (also called evapotranspiration);

*Q*—total runoff.

Evapotranspiration can also be determined using methodological procedures already described in the literature, such as the Thornthwaite method, Penman–Monteith method, and many others [39].

$$E \ = 0.0018 \times (T + 25)^2 \times (100 - r),$$

where

*E*—potential monthly evapotranspiration (mm);

*T*—average monthly air temperature (°C);

*R*—average monthly relative air humidity (%).

I will also mention the universally applicable method of [40], where the calculation for the climatic type of the model area is as follows:

Humid climate:

$$PET \ = 0.013 \times \left[ \frac{T}{T + 15} \right] \times (R + 50)$$

where

*PET*—potential monthly evapotranspiration (mm);

*T*—average monthly air temperature (°C);

*R*—average monthly solar radiation (cal/cm$^2$).

### 3.3.2. Calculation of Runoff by the CN Curve Method

The calculation of runoff by the CN curve method takes input from two basic steps: the assignment of soils in the area to a hydrological group, and the assignment of CN parameters to a specific land use. By overlapping the map of the hydrological groups of the soils with the map of land use, we get individual areas with a numerical CN parameter. The hydrological groups of soils are divided into 4 basic categories (A—soils with a high infiltration rate; B—soils with a medium infiltration rate; C—soils with a low infiltration rate; D—soils with a very low infiltration rate). Land use has a CN number assigned to it in a given hydrological group. High CN values (e.g., 95) indicate areas where most of the precipitation flows off as runoff and the soil has a low water retention capacity. Conversely, low CN values (e.g., 40) indicate areas with high precipitation retention (e.g., forests on deep soils). The method of calculation of runoff by CN curves takes, as the final value of runoff (surface and subsurface), the amount of precipitation that passes through the end of the river basin profile [41–43]. The CN curve method is an empirically obtained procedure for determining the amount of runoff. More details may be found here: https://www.hydrocad.net/curvenumber.htm.

In this methodology, direct runoff is calculated as

$$H_0 = \frac{(H_s - 0.24)^2}{(H_s + 0.8A)}$$

where

$H_0$—direct runoff (mm);

$H_s$—precipitation (mm);

A—potential water retention (mm)—we determine this from the CN curves by the following formula:

$$A = 25.4 \times \left(\frac{1000}{CN} - 10\right)$$

The calculation of the outflow was performed in the ArcMap environment using a raster calculator. The results of the calculation are presented in Table 4 and in a graphical depiction of the monthly LWRC values in the study area (Figure 8).

**Table 4.** Hydrological balance for 2018 in the test area.

| Month | Temperature Average—°C | Precipitation—mm/m$^2$ | Evapotranspiration—mm/m$^2$ | Surface Runoff—mm/m$^2$ | Total Balance |
|---|---|---|---|---|---|
| January | −2.56 | 20.8 | 0 | 0 | 20.8 |
| February | −7.79 | 16 | 0 | 0 | 16 |
| March | −3.55 | 65.8 | 0 | 9.21 | 56.59 |
| April | 9.00 | 38.6 | 16.28 | 24.44 | −2.12 |
| May | 11.40 | 71 | 18.3 | 38.65 | 14.05 |
| June | 13.28 | 162.8 | 42.34 | 78.93 | 41.53 |
| July | 14.83 | 97.6 | 52.96 | 50.32 | −5.68 |
| August | 15.49 | 84.4 | 59.31 | 44.53 | −19.44 |
| September | 10.40 | 114.8 | 48 | 57.87 | 8.93 |
| October | 6.70 | 73.4 | 16 | 39.71 | 17.69 |
| November | 1.11 | 14.4 | 1.3 | 13.81 | −0.71 |
| December | −3.78 | 34.2 | 0 | 0 | 34.2 |
| year | Average Temperature | Total Precipitation | Total Evapotranspiration | Total Surface Runoff | Balance in mm/m$^2$—Whole Year |
| 2018 | 5.50 | 793.8 | 254.49 | 357.47 | LWRC 181.84 |

## 4. Results

### 4.1. Determination of Water Retention of the Soils (SWRC)

In this work, we used a simple algorithm, the result of which is a cumulative index, to evaluate SWRC. Given the fact that the direct measurement of hydrological data on soil is very difficult when it comes to capacities [26], it is common to use statistically expressed pedotransfer functions (PTF) for the indirect estimation of hydrolimits.

These pedotransfer functions are not applicable for the purposes of comprehensive SWRC spatial assessment. For this purpose, we used the spatial representation of the soil cover, using vector databases of soil units, parameters of their quality, and other evaluated attributes, such as skeletality, soil granularity, and morphometric data about the landscape (ABC) (in GIS/ArcMap 10).

The vector database of soil subtypes [35] considers the anthropogenic factor as well as other attributes of soil cover. We deliberately did not use a vector database of rated soil-ecological units (ABC) [35] for our research, as several inaccuracies arose during its creation, which we partially eliminated by using the DMR (10 × 10 m) model. Figure 4 shows the results of syntheses in the form of homogeneous polygons of abiotic complexes, with the numerical coding of all input parameters indicated.

When creating the algorithm, we based it upon the pedotransfer rule. This is based on an assumption that is also confirmed by direct measurements of pF values: the higher the clay fraction percentage in soil, compared to the dust and especially to the sand fractions, the higher the water storage capacity, and thus also the higher the water retention capacity. It is similar for soil depth: the deeper the soil, the more water can accumulate in its profile. Another important factor affecting the soil retention capacity is the morphometric characteristics of the relief, that is, the slope [31].

The range of values of the cumulative SWCR index were divided into 10 categories in a methodology published in a previous study [33]. To evaluate the soil's potential to accumulate water in the soil, we chose the categories for water resources derived from FWC (mm) from the study in [31]. Interpretation of the values of coefficients assigned to the selected soil attributes was carried out using an algorithm, and the projection of the results onto the vector database of ABC polygons is presented in Figure 5.

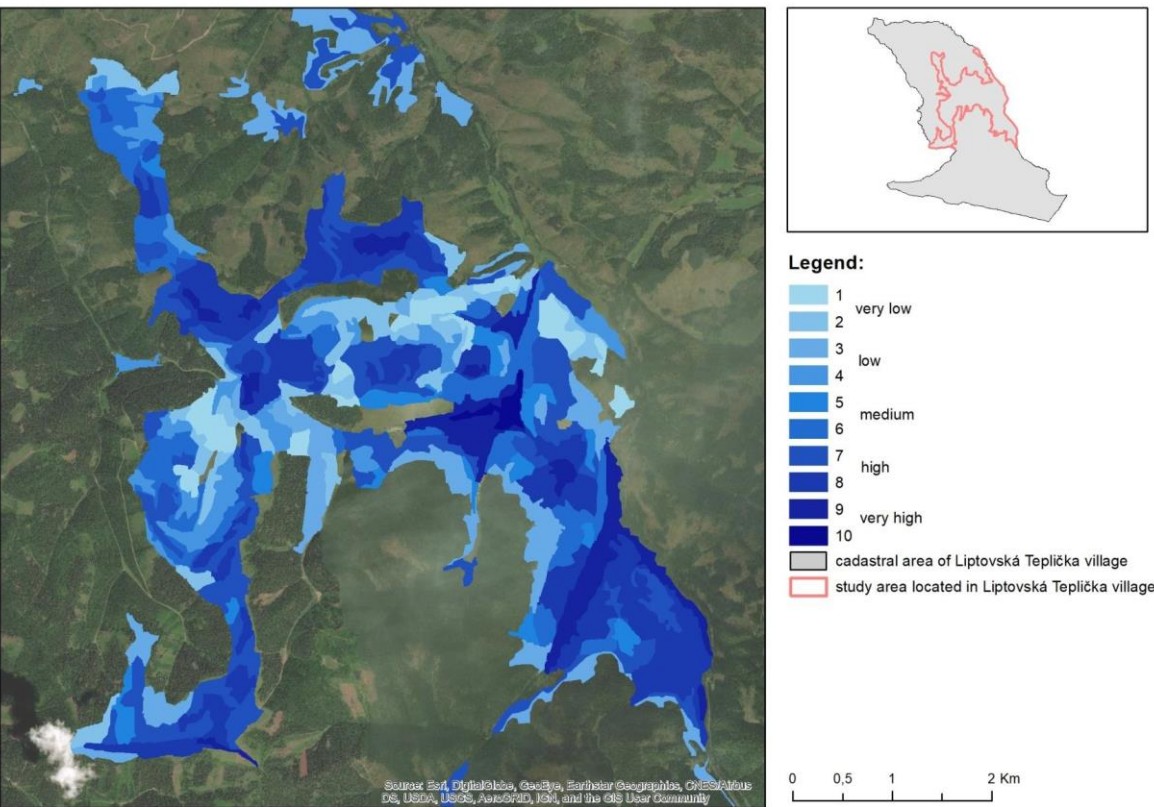

**Figure 5.** Soil water retention capacity (SWRC). Source: Base map, Esri, 2020.

*4.2. Determining the Weighting Coefficient and Calculating the Retention Potential for Types of LSL and Historical Structures of the Agrarian Landscape (HSAL)*

An important phenomenon of the cultural landscape is the presence of historical landscape elements and historical landscape mosaics, which were created in the past by various human activities; some of these are abandoned or nearly so, while others are still in use. They are also referred to as historical structures of agrarian landscapes (HSAL), historical phenomena of landscapes, historical elements of land use, or traditional forms of land use [36].

The overall landscape structure can be expressed mainly by indicators of composition (number of different types of landscape elements) and configuration and spatial characteristics of the elements [44]. The areas of historical structures of the agrarian landscape (HSAL) predominantly display a smaller-scale fragmentation of the landscape, with smaller areas of individual landscape elements, than intensively used agrarian landscape. The traditional agricultural landscape of Liptovská Teplička includes a mosaic of smaller fields of arable land, areas of mowed and otherwise managed meadows, and grazed and abandoned pastures, but also fallow land. The landscape also includes various types of seminatural and nature-friendly habitats, which form various vegetation forms in the landscape. In addition to PGL (abandoned permanent grasslands), there is a lot of non-forest woody vegetation (NWV), which includes drawbridges, riparian vegetation, accompanying vegetation, planted hedges, seepage belts, and abandoned areas overgrown with solitary trees and shrubs (Figure 6). The current use of the landscape is dominated by relatively compact elements of commercial forests, into which the outcrops of flower meadows and pastures are wedged. A mosaic of elements of woody vegetation is formed on the edges of the PGL and the originally cultivated terraced fields. A substantial part of the agricultural land is used in accordance with natural conditions, preserving the unique character of the traditionally used landscape and keeping the elements of the historical landscape structure in good condition.

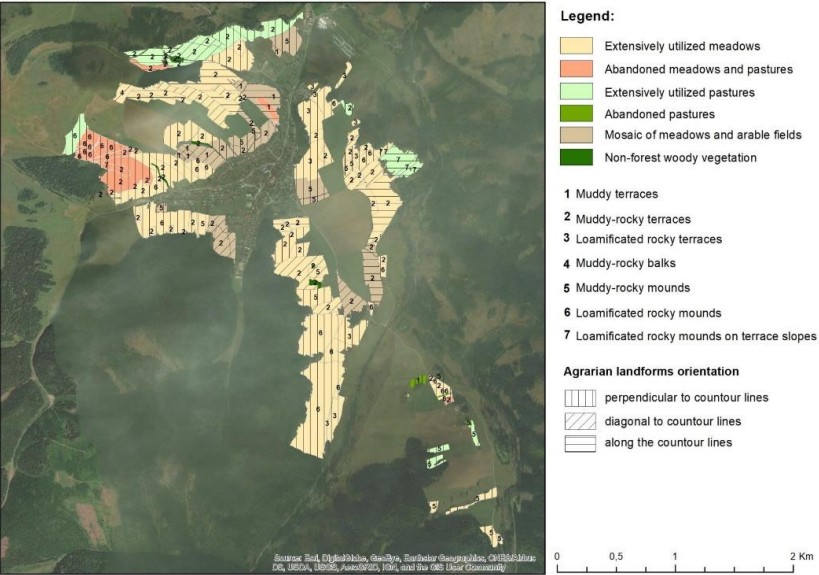

**Figure 6.** Types of historical structures of the agrarian landscape (HSAL), evaluation of the orientation of ramparts and terraces with respect to contour lines, and their classification into categories. Source: Base map, Esri, 2020.

The area belongs to the Váh river basin and to the Čierny Váh sub-basin. On the upper course, the Čierny Váh stream takes in the Holičná stream, with two significant tributaries situated lower down, the Ždiar brook and Teplička. All streams have the type of rain-snow runoff regime with high water content in April to June, the average monthly flow reaching its highest value in May and the lowest in January and February. The average annual flow is around 3 m$^3$ s$^{-1}$.

In the following table (Table 2), we present information on the elements of the current landscape structure, wherein the extent of elements of the SLS is given in both area (ha) and the percentage of the total area, along with the assigned weighting coefficient for estimating the amount of rainwater retained.

We assigned a coefficient to each element of land cover according to the Tužinský algorithm [2], which we adjusted based on field inspections and experience.

### 4.3. Determination of Retention Coefficient for Parcels with Respect to Orientation of Valleys with Respect to the Slope Line (Sloping)

The boundaries of narrow-strip fields usually have the character of terraces or ramparts running along the contour, along the slope, or obliquely. In the past, their water retention effect was also enhanced by the drainage trenches built along such boundaries. In terms of the Water Plan of Slovakia [45], Liptovská Teplička is also an important water management area for the Spiš-Poprad water supply system with groundwater abstraction, and therefore the preservation or renewal of the management of the use of land and natural resources is an important topic. The current form of land use is mainly as meadows and pastures, which is relatively in line with ecological and environmental conditions. However, we were also interested in the orientation of the ramparts and terraces, as we assumed that the outflow of fallen climatic precipitation could slow down or accelerate the outflow of rainwater. It should also be borne in mind that there is almost no non-forest woody vegetation in the meadows or pastures, which would increase the total capacity of retained precipitation water. An overview of the main types of land in the agrarian part is shown in Figure 5. We divided the direction of the ramparts and terraces (or plot orientation) into three categories: 1—perpendicular to contour lines; 2—diagonal to contour lines, 3—along the contour lines. For plots with a value of 1, we assume a higher volume of outflow from the land, and for those with a value of 3, we assume that the ramparts and terraces will retain more water.

The types of historical structures, and evaluation of the orientation of the ramparts and terraces, both of which were considered in the overall evaluation of the landscape water retention capacity (LWRC), are shown in Figure 7.

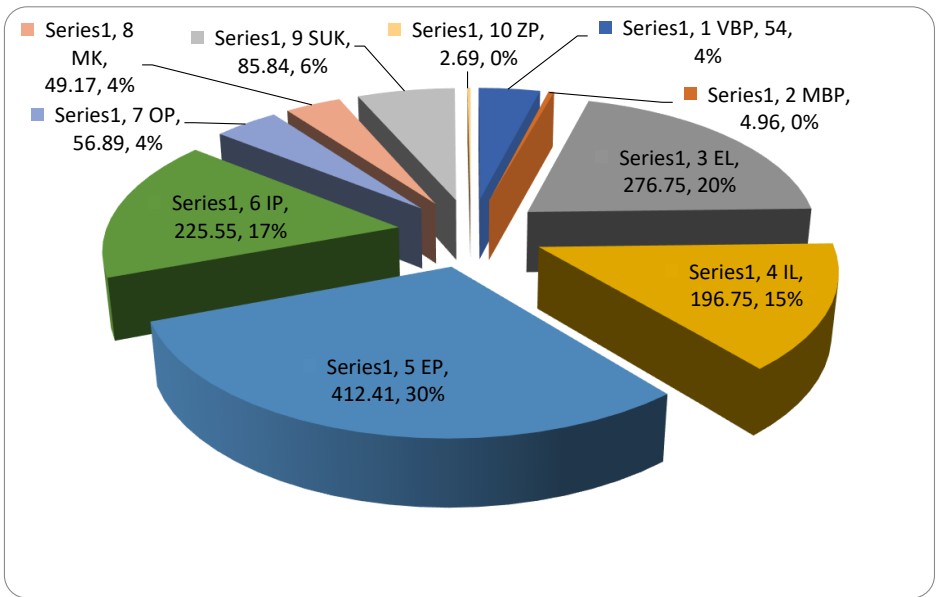

**Figure 7.** Percentage overview of the main types of landscape in the agricultural part of the study area. Legend: SBF—small-block fields, LBF—large-block fields, EM—extensively used meadows, IM—intensively used meadows, EP—extensively used pastures, IP—intensively used pastures, AP—abandoned pastures, MK—mosaics of arable land and PGL, BA—built-up areas, SUC—successive stages of woody plants (covering 40–80% of the area) on abandoned areas, CF—coniferous forests.

## 5. Analysis of LWRC Categories with Identification of Landscape Use and Management—(1812.48 ha—Agrarian Area)

### 5.1. Very Low to Low Landscape Retention (LWRC 1.5–6) (39.91%)

The category of landscapes with a significant number of polygons (and hence a significant area) showing very low to low retention capacity are intensively used pastures (29.33 ha), extensively used meadows (50.40 ha), and extensively used pastures (141.055 ha). Occurrence of HSAL in this LWRC category is rare, and occurs only when the orientation of the ramparts and terraces along the slope allows a faster outflow of rainwater. Within this category, there are also built-up areas where it is assumed that the retention capacity is extremely low. This category also includes a certain amount of large-block fields (58.96 ha). More rarely, there are also home gardens and landscaped areas (4.34 ha). As we start from a synthesis of ABC, landscape cover and orientation of the ramparts, we note that the main soil units (MSU) present in the study area are mostly Skeletic Dystric Cambisol (Loamic, Aric) on flysch rocks on steep slopes, 12–25°, (MSU 82); Skeletic Dystric Cambisol (Loamic, Aric) on flysch rocks, (MSU 78); Rendzic Calcaric Eutric Leptosols (Loamic) on steep slopes, 12–25° (MSU 92), and shallow Rendzic Calcaric Eutric Leptosol (Loamic) (MSU 90). Due to the soil conditions and shallow soil profiles (which have a very low water-holding capacity), and the orientation of the ramparts and terraces, it is the latter that predominantly conditions the very low to low water-holding capacity of the landscape. However, morphological conditions that have been identified in the formation of ABC by DMR are also important factors. In these critical hydric zones, a change in management needs to be proposed. Within the agrarian landscape there is also an isolated forest vegetation of coniferous forests (209.17 ha), which was recorded in four WRL categories (5.2–6.0). Non-forest woody vegetation (NWV), which has significant ecostabilizing effects, occupies a more continuous area only over an area of 18.61 ha. Scattered, smaller areas of NWV are also located in the evaluated locality. The landscape of Liptovská Teplička is also used for recreational purposes: winter sports activities, downhill grassy areas, and cottage settlements (30 ha) predominate here. The total area of agrarian land with very low to low retention is 723.17 ha (39.91%), which is a relatively large area. The area of fields with ramparts and terraces (HSAL) oriented in the direction of the contour lines is 52.87 ha. In this area, which is only a small part of the total, the orientation of the ramparts and terraces along the slope only accelerates the outflow of rainwater (see Table 4).

### 5.2. Mean Landscape Retention (LWRC 6.1–8.2) (43.01%)

The landscape with a medium value of retention is characterized by the predominance of Skeletic Dystric Cambisol (Loamic, Aric) on flysch rocks, medium deep (MSU 78). Rarely, there are Rendzic Calcaric Eutric Leptosol (Loamic) (MSU92) (10.73 ha), used as extensive and intensive meadows. Based on zonal statistics, the land is used mainly in the form of intensive meadows (105.63 ha), with extensive meadows covering an area of 14.06 ha and some extensively used pastures (149.95 ha). These three landscape types dominate this LWRC category. In this category, this is also a relatively small amount of abandoned areas (31.85 ha) with successive stages of woody plants (40% –80% of the area coverage). We also recorded the occurrence of coniferous trees over an area of 8.98 ha. The area of the landscape with medium retention represents, in total, an area of 780.52 ha (43.01%). We also recorded the occurrence of recreational areas and built-up areas in a relatively small area. As part of management measures, we would recommend a significant increase in the share of NWV in areas with an orientation perpendicular to the direction of the slopes. The area of HSAL with orientation of ramparts and terraces in the direction of the slope is 79.28 ha, the area oriented oblique to the slope is 11.22 ha, and the area oriented perpendicular to the slope is 23.40 ha. Again, these are relatively small areas compared with the total area of landscape with a medium LWRC. The medium LWRC areas are predominantly medium–deep soils, with extensive land use in the form of pastures or intensive use as meadows.

## 5.3. High to Very High Retention (LWRC 8.3–13.2) (17.61%)

In this category, the soil types are Skeletic Dystric Cambisol (Loamic, Aric) on a flysch rocks, medium–deep and heavy (MSU 78), and Gleyic Fluvisol (Clayic) (MSU 12), There is also some occurrence of shallow–medium, heavy Rendzic Calcaric Eutric Leptosol (MSU 90) used as extensive pastures (44.56 ha). Abandoned agricultural areas (18.65 ha), extensive pastures (51.35 ha), and extensive meadows (68.89 ha) are found in this category, with relatively large areas of them covered with significant hydric types of land cover, such as cuttings, isolated groups of coniferous forests, groups of trees, bushes, and riparian vegetation of watercourses. There are also landscaped areas, and recreational and sports areas with a permanent lawn. In total, the latter occupy 17.61% of the agrarian land. The built-up area covers 2.69 ha (Figure 8). The HSAL area with ramparts and terraces oriented along the slope covers an area of 25.68 ha, which is a relatively small area, within which there is also an accelerated runoff of rainwater. The area of HSAL with obliquely oriented ramparts and terraces is 38.41 ha. Again, this is a small area, but one which could have a significant effect on the local water dynamics. The HSAL area with ramparts and terraces oriented perpendicularly to the slopes, covering 16.91 ha, is not significant in terms of the total volume of water retained in the study area.

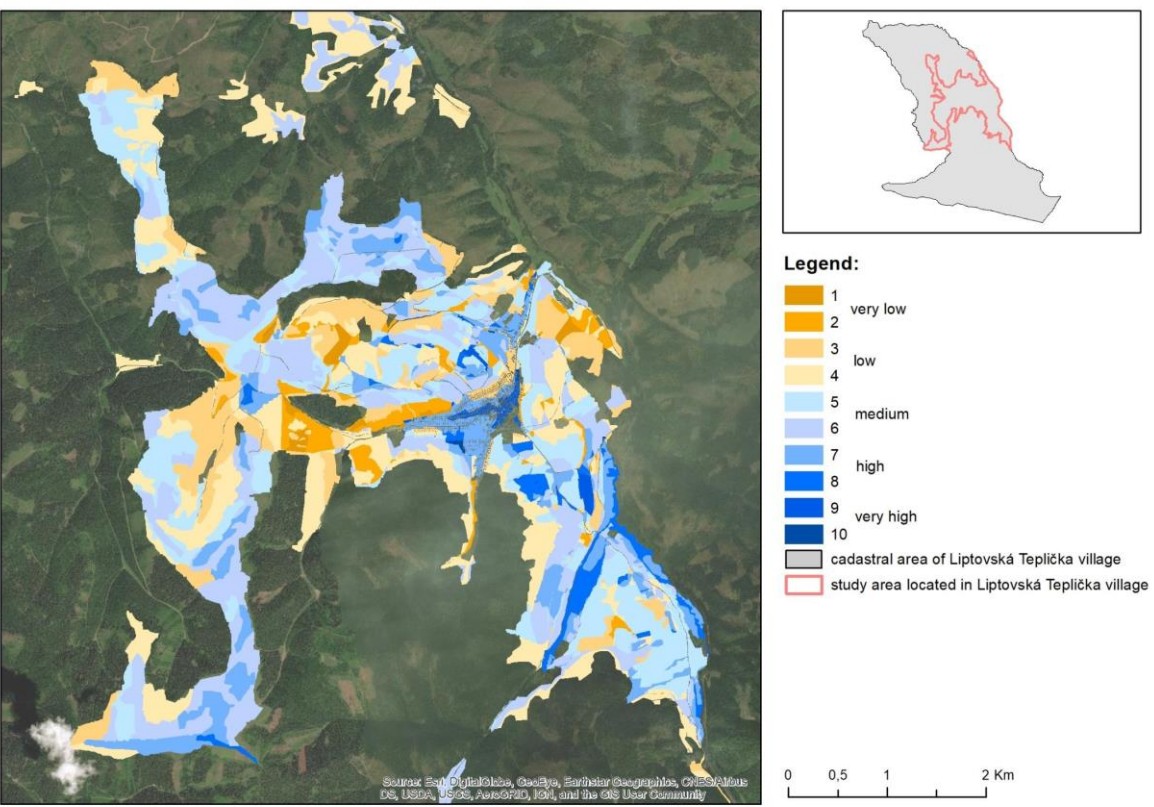

**Figure 8.** Spatial differentiation of LWRC in Liptovská Teplička. Source: Base map, Esri, 2020.

## 6. Calculation of Hydrological Balance in the Area of Interest, Liptovská Teplička

The hydrological balance has been calculated for each month in 2018, but we also present the total balance for the year (Table 4). In our study, we used accurately measured climatic data, and relatively accurate data on evapotranspiration, in conjunction with the current landscape structure and selected attributes of the soil cover. It follows from the above that we obtained a relatively accurate estimate of the volume of water retained in the landscape, as well as other categories such as evapotranspiration and the amount of runoff of surface water from the landscape.

Based on the above calculations from the total rainfall for 2018, the volume of water retained in the land was only 181.84 mm/m$^2$, which means 22.9% of the fallen atmospheric precipitation. Up to

44.9% of the water from the total precipitation drains away in the form of surface and subsurface water. This is a high value, which means that management adjustments are needed in the country, as well as changes in land use in critical hydric zones. Evapotranspiration represents the loss of water by evaporation, which comes to 32.03% of the fallen atmospheric precipitation (see Figure 9).

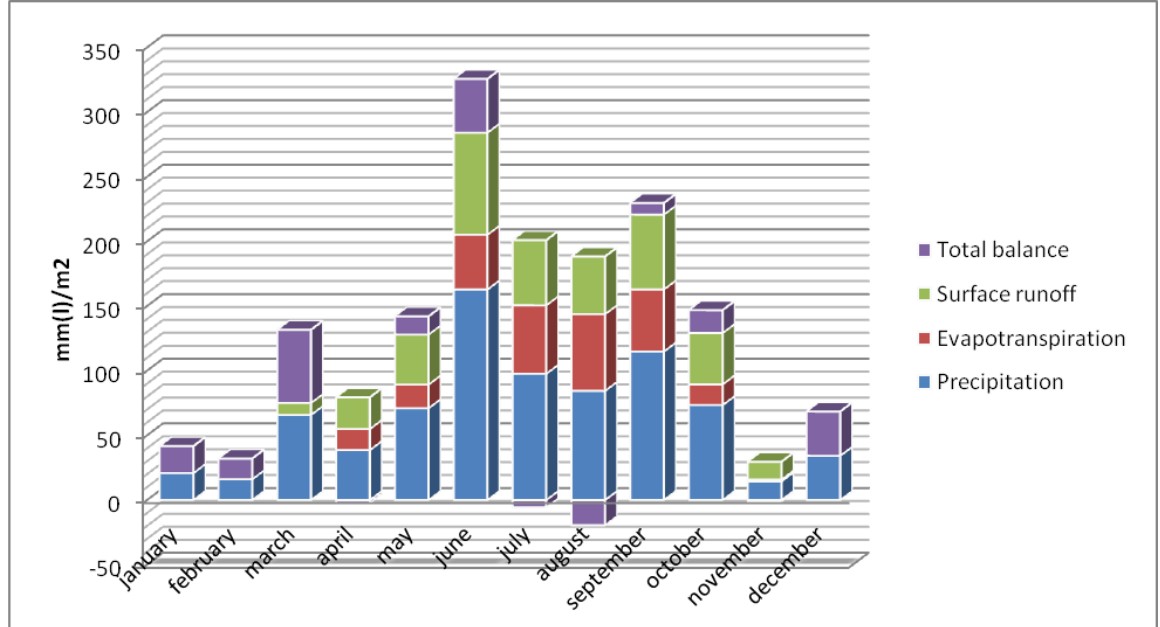

**Figure 9.** Graphical representation of the hydrological balance.

By calculating the hydrological balance (the water regime of the land), it was also possible to derive the actual volumes of water that the country retains for individual LWRC categories. However, the predominance of shallow to medium–deep cambium soils and rendzinas, and land use consisting mainly of meadows and pastures on polygenic hills, conditions the overall high losses of volumes of surface water runoff from atmospheric precipitation. The graphical representation of the hydrological balance also shows that high values are recorded not only in the category of surface runoff but also in the category of evapotranspiration, especially in the growing season. From the above facts, it follows that a more efficient way of land use is needed, and especially that the area of hydric types of land should be significantly increased.

## 7. Discussion

Many authors [46–48] emphasize the importance of land use and its influence on the quantity and quality of water resources. Therefore, in nontechnical methods, one can distinguish planning methods (spatial arrangement) and methods that are dependent on agricultural use. It is relatively hard to evaluate the effect of the landscape morphology and of changes in the land use on the water retention. For these reasons, we used a landscape-ecological approach in our study, where selected detailed attributes of landscape cover, as well as attributes of soil cover, were included in the LWRC assessment. It was the character of the soils and morphology in the test area that conditioned the overall low retention capacity of the SWRC, as well as the low proportion of hydric types of the landscape, which had a significant impact on the LWRC.

Other authors who study water movement in the country [10,11] agree—on the basis of long-term monitoring of small river basins—that local flash floods during high-intensity torrential rainfall in the upper parts of mountain basins with relatively steeps slopes are amplified mainly by the small thickness of the soil profile, with its relatively small retention capacity and the occurrence of a low-permeability rock layer at a depth of less than one meter. After the soil profile is filled, a subsurface and subsequently

a surface runoff ensues, which can cause erosion and floods. These unfavorable (in terms of local flood formation) soil properties cannot be significantly influenced, and thus the management of soil cover properties does not lead to a reduction in the risk of floods. According to the above authors, another important factor is thus the permeability of the geological subsoil.

The expert estimation of the coefficient values in the mathematical equations for the calculation of the LWRC in the methodological part will, of course, not be perfectly accurate and objective. However, the values of the coefficients in the equation were based on the works of authors who have been dealing with this issue for a long time. The coefficient of the SWRC calculation, using the algorithm published by [33], took into account a wide range of evaluated soil attributes (Table 5, Figure 5) for which we assumed a correlation with the natural phenomenon under evaluation. The overall calculation of LWRC values also included coefficients based on a study [2] on the volume of water retained by an SLS. Categories with a lower LWRC value of 1.5–6 have a low potential for retaining water in the landscape. LWRC with a value of 8.3–13.2, on the other hand, has a high potential to retain water. We also took into account the orientation of the ramparts and terraces with respect to the contour lines; however, the effect of this factor had a negligible effect on the overall LWRC assessment due to the small areas involved.

**Table 5.** Overview occurrence of the SWRC categories (Liptovská Teplička).

| Soil Water Retention Capacity (SWRC) | | | | | |
|---|---|---|---|---|---|
| Category of SWRC | Degree of Water Retention Capacity (WRC) | Area (m $^2$) | Area (ha) | Area (%) | Category of Water Resources (Resources Derived from FWC) |
| Very Low | 1 | 769,260.57 | 76.93 | 4.26 | |
| | 2 | 1,389,818.47 | 138.98 | 7.70 | (≤50 mm) |
| Low | 3 | 3,128,321.25 | 312.83 | 17.33 | |
| | 4 | 1,169,236.12 | 116.92 | 6.48 | |
| Medium | 5 | 800,037.31 | 80.00 | 4.43 | (51–100 mm) |
| | 6 | 1,830,055.31 | 183.01 | 10.14 | |
| High | 7 | 3,655,225.27 | 365.52 | 20.25 | |
| | 8 | 3,909,126.74 | 390.91 | 21.66 | ≥100 mm |
| Very high | 9 | 1,281,727.29 | 128.17 | 7.10 | |
| | 10 | 121,210.15 | 12.12 | 0.67 | |
| Total | | 18,124,818.48 | 1812.48 | 100 | |

Legend: FWC—categorization of water resources in terms of the full water capacity of the soil profile.

In our study, we also used the method of CN curves (with appropriate modifications for local conditions), which is a generally used procedure for calculating the volume of runoff from largely unmapped river basins without needing hydrological field research [41–43]. Its main advantages are speed and ease and versatility of use. The accuracy of the output is mainly ensured by high-quality and up-to-date input data (soil, land cover, and land use maps) and correct classification of individual hydrological groups of soils and of land use. Different sources also give slightly different CN values during classification, thus necessitating expert estimation, which is one of the disadvantages of using this method.

## 8. Conclusions

In this paper, we have indicated one of the possible methods of evaluating the landscape potential water capacity (LWRC) of the country, which follows the method of evaluating the potential water capacity of the soil (SWRC) that we discussed earlier [33].

We planned this study making the assumption that the mountain landscape with preserved elements of historical agricultural use contributes to a favorable hydrological balance. We consider geomorphological, climatic, soil, and hydrogeological conditions to be decisive factors that determine the distribution of land cover elements and land use elements. The model area has a specific location near the regional watershed, and at the same time represents a significant tributary of the Čierny Váh below its spring area at an altitude of 1800 to 1900 m. Due to the representation of relatively shallow soils, the relief dynamics, and the high total precipitation, it is necessary that soil protection elements

and hydrically effective types of land cover be stabilized as a priority on sloping areas in the areas of watersheds, springs, and watercourses, which characterize the study area of Liptovská Teplička.

From the results of calculating the hydrological balance, it was also possible to derive real volumes of retained water for the specified categories, which are listed in Table 2. We found overall high losses of volumes of runoff surface water from atmospheric precipitation. This is also confirmed by the results of zonal statistics and hydrological balance, which are more or less correlated. Our prediction of the land's ability to retain water has not been confirmed, even with the traditional narrow strip fields present in the area and a high proportion of the meadow and pasture forms of land use. One of the reasons for the unfavorable water balance in the area is the predominance of shallow to medium–deep Skeletic Dystric Cambisol (Loamic, Aric) on flysch rocks, and Rendzic Calcaric Eutric Leptosol (Loamic), with lower soil retention capacity, but the uneven rainfall distribution is probably also a contributing factor, with intense rainfalls in which much of the water flows from the slopes and also from the basin in the form of surface runoff.

An integrated approach to flood prevention and protection is becoming increasingly important, and the management of this issue can be co-determined by the social, economic, and environmental limits of river basin use while meeting society's needs. A combination of various targeted preventative technical and nontechnical measures can to some extent mitigate the negative consequences of extreme flood events on society (e.g., best practices on flood prevention, protection, and mitigation; flood risk management in the form of flood prevention, protection, and mitigation [44,45]). Such flood protection measures may be based, in particular, on the following principles: 1. on increasing the natural retention capacity of the area, for example, changing the land use patterns and land management, which can also be done in coordination with erosion protection; 2. on the modification and renewal of watercourse beds while respecting the principles of riverbed stabilization and with emphasis on its revitalization; 3. on increasing the natural retention volumes in river basins and flood and floodplains; 4. on increasing the retention capacity of urbanized areas by building green infrastructure and natural retention areas.

This study could contribute to the incorporation of proposals and measures into LEPs, and to the specification and supplementation of regulations connected to spatial plans at the local level. Inclusion of environmental regulations in the spatial plans is necessary for ensuring the protection of natural resources and optimal spatial functional development.

Landscape ecological plans with incorporation of proposals and regulations can serve as a basis for the allocation of natural floodplains, where any human activity should be excluded (especially urbanization activities and the like) and no houses should be built, and where rivers should be given more space by expanding floodplain areas outside municipalities.

At the time of preparation of the study, as well as other studies in the model area of Liptovská Teplička, anti-erosion measures were proposed, which are clearly designed to reduce surface wash and runoff of the topsoil. These proposals were incorporated as measures within the landscape ecological plans as well as the landscape revitalization program. The aim of the Program of Landscape Revitalization and Integrated River Basin Management of the Slovak Republic, approved by the Government in October 2010, is to create and activate long-term conditions not only for effective operation of a comprehensive and integrated system of flood prevention measures and water management in the country, but also for protection of the soil fund, including measures aimed at reducing the risks of soil erosion and increasing the country's resilience to extremes and changes in weather and climate. Other appropriate measures include sowing grass or afforesting parts of the river basin, increasing the proportion of NWV, naturalizing channelized parts of streams, and building polders.

**Author Contributions:** Writing: Original draft—Z.K., P.K., D.K., and J.H.; data curation and investigation—Z.K., P.K., D.K., and M.D.; funding acquisition—J.H. and M.D.; supervision—Z.K.; writing: Review and Editing—J.H. and Z.K. All authors have read and agreed to the published version of the manuscript.

**Funding:** This research was supported by the project APVV-17-0377: Evaluation of modern changes and development trends in the agricultural landscape of Slovakia (2018–2022), and project GP VEGA 2/0078/18: Research into the biocultural values of the landscape (2018–2021).

**Acknowledgments:** The authors would like to thank the editors and reviewers for the useful comments, which greatly helped in improving the manuscript. We also thank Mathew Sebastian and James R. Asher for English correction.

**Conflicts of Interest:** The authors declare no conflict of interest.

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
