# Peer review of "Assessment of Landscape Retention Water Capacity and Hydrological Balance in Traditional Agricultural Landscape (Model Area Liptovská Teplička Settlements, Slovakia)"

_water, doi:10.3390/w12123591_

Round 1
Reviewer 1 Report
All my comments are corrected in the new version of the submited manuscript.Author Response
Thanks for your work.
Reviewer 2 Report
The article should have been delivered with track changes and it was not what makes the review difficult. Although, for what I saw the article should be accepted with minor changes. Small corrections are suggested in green in the text enclosed.

Author Response
Correct 2. review
Reply to point L 107
we present the following literature in the text:
Karina VR Schafer, Ram Oren, Chu-Tal Lai, Gabriel G. Katul., Hydrologic balance in an intact temperate forest ecosystem under ambient and elevated atmospheric CO2 concentration. First published. Global, Change Biology, Vol., 8, Issue 9, 2002
Gahala, A.M., Seal, R.R., and Piatak, N.M., 2019, Hydrologic balance, water quality, chemical-mass balance, and geochemical modeling of hyperalkaline ponds at Big Marsh, Chicago, Illinois, 2016–17: U.S. Geological Survey Scientific Investigations Report 2019–5078, 31 p., https://doi.org/10.3133/sir20195078. ISSN: 2328-0328
Reply to point L111
We have supplemented the following literature:
Rožňovský, J., Chuchma, F., Fiala, R. 2018. Základní vlahové bilance na území ČR v suchých letech. In Vlyv antropogénnej činnosti na vodný režim nížinného územia: Zborník recenzovaných príspevkov - Proceedings of peer-reviewed contributions [elektronický zdroj]. - Bratislava; Michalovce: Ústav hydrológie SAV: Výskumná hydrologická základňa, 2018, s. 223-232. ISBN 978 – 80 – 89139 – 41 - 5.
Šoltész, A., Baroková, D. 2011. Impact of landscape and water management in Slovak part of the Medzibodrožie region on groundwater level regime. In Journal of Landscape Management. 2011, vol. 2, no. 2, pp. 41-45
Reply to L113.
In the text we have specified these sources and and we also list the original documents of Slovak Government:
Caims, Jr. J., Crawford, T. V., Salwasser, H. (eds.): Implementing Integrated Environmental Management. Blacksburg : Virginia Polytechnic Institute and State University, 1994.
Hrnčiarová, T. a kol.: Metodický postup ekologicky optimálneho využívania územia v rámci prieskumov a rozborov pre územný plán obce. Bratislava : MŽP SR, Združenie KRAJINA 21, 2000, 136s.
Lehotský, M.: Riečna krajina a jej udržateľný rozvoj – nová oblasť aplikácie integrovaného prístupu. In: Izakovičová, Z. (ed.): Integrovaný manažment krajiny – základný nástroj implementácie trvalo udržateľného rozvoja. Bratislava : ÚKE SAV, MŽP SR, SVS, 2006, s. 155 – 159 .
Druhý realizačný projekt Programu revitalizácie krajiny a integrovaného manažmentu povodí SR 2011 (materiál schválený uznesením vlády SR č. 590/2011 dňa 7.9.2011
DOUROJANNI, A., JOURAVLEV, A., 1995: Integrated River Basin Management in Latin America. In Domenica M. F. (ed.) Integrated Water Resources Planning for the 21st Century. New York: p. 953–956
KUNDRÁT, T., PROSBA, J., KUNÍKOVÁ, E., BÖDI, L., DOBROTKA, S., KAVEČANSKÝ, Š., NOVOSAD, M., SESZTÁK, J., ONDREJČOVÁ, H., VÁGAŠIOVÁ, M., VINDIŠOVÁ, D., BORGULOVÁ, B., HOŠNOVÁ, V., MATO, J., ROMČÁK, P., PIETERSEN, W., OORT, VAN J. D., KOSTER, R., WAVEREN, VAN H., URBELS, A., SLUIS, VAN J. W., LUEVERMAN, J., 2002: Integrovaný vodohospodársky plán povodia Hornádu. Pilotná štúdia. Matra 2002: Implementácia Rámcovej smernice EÚ pre vodu SR. SVP-PBaH, VÚVH, VVaK, SHMÚ, Bratislava: 125 p.
Program revitalizácie krajiny a integrovaného manažmentu povodí SR a návrh jeho realizačného projektu 2010 (materiál schválený uznesením vlády SR č. 744/2010 dňa 27.10.2010)
Prvý realizačný projekt Programu revitalizácie krajiny a integrovaného manažmentu povodí SR 2011 (materiál schválený uznesením vlády SR č. 183/2011 dňa 9.3.2011)
Reply to L 128
The connection of the landscape-ecological approach (based on the LANDEP methodology) with the hydrological balance was used for the first time. The result of landscape ecological outputs is the determination of the country's potential for water retention capacity, but realistic estimates of the volume of water retained in the country cannot be determined. For this reason, we used an additional methodology of hydrological balance, where both the attributes of land and land use enter, but in a very general level, the focus is on measured climatic data, evapotranspiration, from which it is possible to derive real volumes of retained water within the allocated landscape-ecological complexes. (LEC). The hydrology itself, which has so far used climatic data, evapotranspiration and flow in the watercourse, is gradually developing and using the attributes of the soil and landscape, the so-called ecohydrology.
At the time of preparation of the study, as well as other studies in the model area of Liptovská Teplička, anti-erosion measures were proposed, which are clearly designed to reduce surface wash and runoff of the topsoil.These proposals were incorporated as measures within the Landscape Ecological Plans as well as the Landscape Revitalization Program. The aim of the Program of Landscape Revitalization and Integrated River Basin Management of the Slovak Republic, approved by the Government in October 2010, is to create, activate and create long-term conditions not only for effective operation of a comprehensive and integrated system of flood prevention measures water management in the country, protection of the soil fund, including measures aimed at reducing the risks of soil erosion and increasing the country's resilience to extremes and changes in weather and climate.

This manuscript is a resubmission of an earlier submission. The following is a list of the peer review reports and author responses from that submission.
Round 1
Reviewer 1 Report
General comments: The article is quite well written, interesting and valuable material, only minor corrections are necessary. I have attached my comments in the proofreading. I recommend publishing after the appropriate corrections.
L 39-40
Surface runoff is a natural part of natural processes. The definition is realistic only for large territorial units.
Fig 2
Average temperatures are not suitable as columns. It is more appropriate to express temperature as points. Temperature is not a summary characteristic, such as precipitation.
L 211
Define the formula (mathematically) of SWRC. Does the calculation respect hydro limits?
L 275
Were PTFs estimated for Slovak soils?
L 306
Unify, modify LWRC categories according to Tab. 3, column 1. Low retention (retention class 3,4) is in Tab 3. 30-70mm. In Table 1, the value for the same categories (3,4) is 0.25m3 / 1m2.
0.25m3 / 1m2 = 0.25m = 250mm !!!!
L 183, 306, 309
In hydrological practice (precipitation, evaporation, water retention in the soil …) units (mm) are used as standard.
m3 / 1m2 = (m) or (mm)
Tab 4
It is not necessary to state the unit (mm / m2), it should be correct (mm)
L638, 654. 656
author's name, font size

Reviewer 2 Report
The authors investigated landscape water retention capacity in Slovak. They developed several indicators, such as soil water retention capacity, and landscape retention water capacity. Then they assessed the cadastral area. While the subject is relevant and interesting, this paper is very difficult to follow up. The reviewer has many concerns that needs to be addressed. Therefore, the reviewer cannot recommend it as it currently stands. My detailed comments are listed below:
- This paper is tedious. There are many repeated words. For example, in ln13, two landscapes were used in the sentence. Ln95-97, two where were used in the sentence. In ln145, methodological is the same meaning as approach.
- There are many inconsistences. For example, ln17, they defined LWRC as the Landscape Retention Water Capacity. Then ln27, they defined it as Landscape Water Retention Capacity. Ln 38, they called it as water retention landscapes. Ln104-105, they changed it as the total retention capacity of the landscape. Ln110, LWRC became the potential of water retention capacity. Similarly, SWRC is also inconsistent in the text.
- Any abbreviation should be spelled out in the first time. Then the abbreviation should be used in all the text after that.
- There are grammar problems. This paper needs to be proofread and edited by native English. For example, ln13-14, the sentence is awkward. Ln14-16, check the grammar of the sentence. The third sentence in Abstract has grammar problem too.
- In Section 3.1, SWRC should be defined before you used it. How did you calculate it?
- In Section, Eq. (1) is unclear how to calculate LWRC. Although three terms were provided on the right side of Eq. (1), it is unclear how each term was calculated. All three parameters had not been defined.
- In Section 3.9, it is unclear what was the CN curves. How die you define/calculate the CN parameter?
- In Section 4, the authors ranked SWRC as 4 levels: 1, 2, 3, and 4. However, it is unclear how the levels were ranked. In addition, there are many new concepts, such as historical landscape structures, NWV, PGL and so on. You should present your classifications of land uses in Methodology section.
- Figure 5 is unclear. The reviewer cannot read words in Legend.
- Conclusions are what you deliver to readers. What are your findings? The current conclusions are not appreciated. You need to quantify the spatial and temporal soil water retention capacity. How will the readers use your results?
Reviewer 3 Report
The article deals with a subject that is poorly dealt with at landscape level and should, as the authors suggest, be integrated into spatial planning. However, scientific methodology and writing skills are not adequately applied in the paper in the sense that the relevant state of the art, knowledge gap, hypotheses formulation, research relevance and objectives are poorly developed, explained and placed in the appropriate sections of the article.
There is no specific section for literature review, and the scarce literature references in the introduction are not enough to understand the subject matter and its scientific context. It seems that some general literature in water and soil conservation should be referred as well more specific literature about methodologies for determining water retention capacity at landscape scale. The authors place relevant literature review in the discussion section, where only should be done the explanation of how results contribute to state of the art. Also, discussion should address data and methodology limitations. There are no suggestions for further work or improvements.
In what regards the research relevance, the author’s intention is that results contribute to spatial planning. As such, there should be a description of how the country's planning system deals with these issues, i.e. whether there are regulations establishing natural constraints related to differentiated terrain morphology to be considered in spatial planning, and its importance for the proper functioning of the water cycle at landscape scale. This information should be placed briefly in the introduction, to explain the relevance of the study, and developed in section 2. Study area.
Extensive English language corrections need to be done. Understanding the text is very difficult, the sentences are very long and the discussion of the results is cumbersome. There is also an overuse of abbreviations. Adding to this, even for the concepts that are scientific terminology there seem to be translation errors. For instance “capacity of retained water in the soil profile”. Shouldn’t it be water retention capacity of the soil profile? And “Landscape Retention Water Capacity (LWRC)” that appears in L210 as “LWRC– total water retention capacity of the landscape”, shouldn’t it be Landscape Water Retention Capacity (LWRC)? And what about “potential of soil retention capacity (SWRC)”, in L272 is it like this or as show in L270 “water retention of the soils (SWRC)”? Shouldn’t it be “soil water retention capacity” as it is in Figure 6. Soil water retention capacity (SWRC)? In scopus database there are 138 results for TITLE-ABS-KEY ("soil water retention capacity" )… Correct terminology in English language should be selected and used.
For these reasons, and considering that these are major flaws, I recommend the paper can be considered after major revisions if, and only, these revisions are done properly.
List of changes and errors:
L 48. “[3] have, however, pointed out that” – It is important to refer to authors in addition to the reference number. There are other examples in the article of this that should be corrected
L88. In what regards the “Landscape ecological approach” which is advocated, can you please give a definition and literature for it?
L81-84. Very long sentence for EN language. This is just one example of many in the article.
L88-94. About the Hydrological balance, the authors should give some literature references about this concept and methodology.
L97-100. “Therefore, it is necessary to protect this zone with a predominantly high retention capacity, either in the form of legislation, well-established regulations and measures in many of the above-mentioned strategic documents, which also include spatial plans with a developed landscape and ecological plan.”
I completely agree with this normative position but the sentence is too long and difficult to understand. I recommend that you rephrase it into 2-3 sentences. Which are the “above-mentioned strategic documents”? They should be referred again so that we don’t have any doubts. Do the referred documents establish the need for “spatial plans with a developed landscape and ecological plan”? Or is this a suggestion from the authors? What is a “landscape and ecological plan”?
The article should also be reviewed for general typo errors: Examples: L137 figure 2.and 3.; L177 “capacity (SWRC) ) assessment”
Figure 2. should be corrected since the legend should not be in front of the chart area
L159. “use of land”− Do you mean Land use and Land Cover data?
L175. What is DMR? Abbreviations should be adequately described before being used.
L178. “In the first step of the methodological procedure, we created abiotic complexes (ABC) by synthesis of vector analytical databases.” Which spatial analysis tool was used for this synthesis?
L212,213 what is WR?
L291. Legend to Figure 5. Is not legible
L298. “The range of intervals for individual categories of the cumulative SWCR index was divided into 10 categories in a methodology published in the study of the authors [12].”
Even if the authors base their method in other paper this article should be enough to understand the methodology and to reproduce it. As such this not properly explained.
L322. PGL the abbreviation term is not explained before, it only appears in L327 permanent grasslands (PGL)
L325. “commercial forests” − Production Forests?
Maybe Figures 7 and 8 should be merged? One is the spatial representation and the other the analysis of the Proportion of landscape structures? If you are speaking of different things maybe figure 7 should have a map with the main types of agricultural landscape.
L372-375. “4.4. Determination categories LWRC with and identification use of landscape and management - (1 812.48 ha - agrarian area) Very low to low landscape retention (LWRC 1.5 - 6) (39.91%)” Is this a subsection title??
Figure 9. Spatial differentiation of LWRC. Please don’t use abbreviations in captions. Why do you have two subclasses and colours four the same map class (1-2 very low LWRC)? You should reclassify this and the same goes for the map in figure 5.
L461-463 There should be a map of critical hydric zones. Otherwise your results cannot be used for spatial planning or land use planning.
L465-478 This is literature review that supports your work. It should be placed in a specific section previous to methodology and results. In the discussion you should refer how your results contribute to the existing literature review. You cannot refer literature for the first time here that is not previously commented in a literature review section.
L478-482 The knowledge/methodology gap that your work is trying to solve should be adequately highlighted previously.
L528. “which we discussed earlier [12]” Conclusions do not have references.
L529-530. “Our intention is based on the assumption that the mountain landscape with preserved elements of historical agricultural use contributes to a favourable hydrological balance.” Is this you hypothesis? I so, it should be placed in the introduction or in the Literature review/state of the art section right after the knowledge gap.
L539 No reference to table numbers and figures should be done in the conclusion